# Understanding In-Person and Online Exercise Oncology Programme Delivery: A Mixed-Methods Approach to Participant Perspectives

Delaney Duchek [1,*], Meghan H. McDonough [1], William Bridel [1], Margaret L. McNeely [2,3,4] and S. Nicole Culos-Reed [1,5,6]

1   Faculty of Kinesiology, University of Calgary, Calgary, AB T2N 1N4, Canada; meghan.mcdonough@ucalgary.ca (M.H.M.); william.bridel@ucalgary.ca (W.B.); nculosre@ucalgary.ca (S.N.C.-R.)
2   Department of Physical Therapy, University of Alberta, Edmonton, AB T6G 2G4, Canada; mmcneely@ualberta.ca
3   Department of Oncology, University of Alberta, Edmonton, AB T6G 1Z2, Canada
4   Supportive Care, Cancer Care Alberta, Edmonton, AB T5J 3E4, Canada
5   Department of Oncology, Cumming School of Medicine, University of Calgary, Calgary, AB T2N 4N1, Canada
6   Department of Psychosocial Resources, Tom Baker Cancer Centre, Cancer Care, Alberta Health Services, Calgary, AB T2N 4N2, Canada
*   Correspondence: delaney.duchek@ucalgary.ca; Tel.: +1-(604)-834-9507

**Abstract:** Alberta Cancer Exercise (ACE) is an exercise oncology programme that transitioned from in-person to online delivery during COVID-19. The purpose of this work was to understand participants' experiences in both delivery modes. Specifically, survivors' exercise facilitators and barriers, delivery mode preference, and experience with programme elements targeting behaviour change were gathered. A retrospective cohort design using explanatory sequential mixed methods was used. Briefly, 57 participants completed a survey, and 19 subsequent, optional interviews were conducted. Most participants indicated preferring in-person programmes (58%), followed by online (32%), and no preference (10%). There were significantly fewer barriers to (i.e., commute time) ($p < 0.01$), but also fewer facilitators of (i.e., social support) ($p < 0.01$), exercising using the online programme. Four themes were generated from the qualitative data surrounding participant experiences in both delivery modes. Key differences in barriers and facilitators highlighted a more convenient experience online relative to a more socially supportive environment in-person. For future work that includes solely online delivery, focusing on building social support and a sense of community will be critical to optimising programme benefits. Beyond the COVID-19 pandemic, results of this research will remain relevant as we aim to increase the reach of online exercise oncology programming to more underserved populations of individuals living with cancer.

**Keywords:** exercise oncology; telehealth; synchronous delivery; supervised exercise; group-based exercise

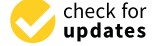



## 1. Introduction

The benefits of exercise in cancer populations are well-documented. Exercise improves aerobic capacity, strength, body composition, mental and emotional health, and quality of life (QOL) [1–6]. Exercise can also mitigate treatment-related side effects such as pain and cancer-related fatigue, improve chemotherapy completion rates, and aid in preventing secondary cancer recurrence or mortality in certain cancers [5,7–15]. Despite these well-established findings reinforcing the health benefits of exercise, only approximately one-third of survivors meet the current exercise recommendations for cancer [5,15]. With few people living with and beyond cancer exercising, research has evaluated the barriers to and facilitators of exercise. Barriers are factors that hinder an individual from exercising, while

facilitators are factors that help an individual to engage in exercise. A systematic review conducted by Clifford et al. (2018) [16] showed that the most common exercise barriers tend to be persistent treatment-related side effects, a lack of time, and fatigue [16]. This same systematic review found the most common facilitators of exercise to be experiencing a feeling of control over health, managing emotions and well-being, improving physical health, and the social benefits of exercising. Similarly, Blaney et al. (2013) [17] found common barriers to be pain, a lack of motivation, weather extremes, travel and time commitments, and costs. Facilitators included managing fatigue, improving QOL, and experiencing a sense of achievement from exercising [17]. Given the demonstrated barriers and low activity levels, there is a need to implement accessible exercise programmes that build healthy exercise habits in cancer populations [18]. COVID-19 was a barrier that interrupted the offering of such exercise oncology programmes, forcing many such programmes to transition to an online platform.

From the onset of the COVID-19 pandemic, a broad expansion of telehealth technology has occurred to deliver remote healthcare to cancer populations. Telehealth broadly refers to providing distance-based exercise or other health-based interventions by utilising communication technologies [19,20]. Rapid improvements over the last ten years in telehealth technology, coupled with the need for online support due to COVID-19-related restrictions, have created an opportunity to optimize remotely delivered supportive cancer care resources [21–23]. Given the barriers that COVID-19 presented to people living with cancer maintaining adequate physical activity levels and quality of life [24,25], there is a need to evaluate supervised telehealth interventions that include videoconferencing to replicate in-person programming within exercise oncology [19].

To our knowledge, there has been very little intervention research investigating synchronously delivered, online, supervised, and group-based exercise oncology programmes. In the last six years, reviews have evaluated the feasibility and effectiveness of telehealth exercise oncology interventions, both before and during COVID-19 [1,26–33]. The results have been predominantly positive in terms of participant acceptability; trial feasibility; and effects on moderate-to-vigorous physical activity (MVPA) levels, quality of life, and self-reported fatigue across a range of cancer diagnoses [23,26,27,32–38]. Two reviews have compared the use of synchronous, asynchronous, or combined exercise oncology telehealth interventions in a home environment, both concluding that there was insufficient evidence to determine which delivery mode is more effective at promoting beneficial health outcomes [28,39]. However, a recent review conducted by Gonzalo et al. (2022) [39] concluded that virtually supervised, group-based exercise oncology programmes were superior to self-directed or unsupervised programmes when looking to maximize health benefits, adherence, and safety [39]. Despite this conclusion, challenges still existed with this delivery mode style, including accessibility, remote exercise testing, and further ensuring participant safety.

In 2021, Dennett et al. [40] used a mixed-methods approach to evaluate the feasibility, safety, and adherence to exercise in a telerehabilitation programme for people living with and beyond cancer. Surveys were used with participants, but interviews were only conducted with staff members; thus, qualitative perspectives of people living with and beyond cancer on the benefits and challenges of both in-person and online exercise oncology programmes are lacking to date [40]. Given this previous research, further evaluation using a mixed-methods approach to understand online, synchronously delivered, group-based exercise oncology programmes is still needed. An opportunity to evaluate such a programme arose at the onset of the COVID-19 pandemic when the Alberta Cancer Exercise (ACE) programme was forced to transition from in-person to online offerings.

ACE is a hybrid implementation–effectiveness study [41] that offers a free, supervised, group-based exercise oncology programme to people living with and beyond cancer over the age of 18 with any form of cancer, up to 3 years post-treatment completion. ACE is led by qualified exercise professionals [42], and delivered to individuals living with and beyond cancer. Qualified exercise professionals are trained specifically in exercise oncology

and how to tailor various exercises to varying degrees of skill and cancer complications during class time [42]. ACE is multimodal and progressive, with exercise options always provided to ensure participants can tailor their movement to meet their needs. ACE was offered at multiple locations and with varying time offerings across Alberta. Classes are 60 min in length, follow a circuit-style design, focusing on aerobic, strength, balance, and flexibility training, and are offered twice weekly for a period of 12 weeks. ACE classes are offered at two levels: baseline and maintenance classes. Baseline classes are the study intervention, are free and the first ACE programme in which participants enrol. ACE maintenance classes are a pay-for-service programme and are accessible for anyone who has completed the initial baseline programme. The programme is delivered across cancer centres in Alberta using established relationships with healthcare workers, print advertising material, and by word of mouth.

ACE baseline evaluation has been reported [41]. Briefly, participant-reported outcomes (PROs) include QOL (physical, social, emotional, and functional well-being), fatigue, and current exercise levels and intention. Physical functioning assessments include lower body endurance, balance, aerobic endurance, lower body flexibility, and shoulder range of motion. Finally, satisfaction with the ACE programme is collected via self-report. PROs are collected at baseline, 12 weeks, 24 weeks, and 1 year, and with ongoing reporting of PROs on a yearly basis until 5 years post-programme completion. Physical functioning assessments are conducted at baseline, 12 weeks, 24 weeks, and 1 year. The ACE programme has demonstrated immense success across Alberta, with over 2600 participants completing the baseline programme to date (2023). Implementation success includes an established clinic-to-community based model with implementation in more than 18 sites across Alberta.

When the ACE programme transitioned to an online platform in April 2020, there was relatively little direction or guidance due to the lack of previously offered exercise oncology programmes that were synchronously delivered, group-based, and supervised via an online format. Synchronous interventions are defined as real-time, face-to-face interactions between the participant and intervention leader (i.e., healthcare provider or exercise oncology instructor) using any technology that permits such interactions. ACE online delivery was built on Zoom, a videoconferencing platform accessible from any mobile or desktop device with an internet or cellular connection. For ACE, all security precautions were taken to prevent the misinformed sharing of personal information of ACE participants, including utilising a password, a secure Zoom link, and the waiting room function to screen participants who were allowed to enter the Zoom room. Participants were sent an instruction guide for Zoom use prior to beginning any online classes or physical functioning assessments.

Many aspects of the in-person ACE programme were transitioned to the online Zoom platform, including the group-based nature of the programme, circuit-style training, and the length and frequency of the programme. Other aspects inherent to the previously established ACE programme had to be adjusted to the new online environment, including how the physical functioning assessments were conducted, the lack of/limited fitness equipment, and class size. Participants were advised that they could use any fitness equipment that they currently owned during the classes, but that no additional equipment was required. This was carried out in an effort to minimize any financial barriers to participants who did not want to purchase their own exercise equipment.

ACE aims to support sustainable exercise habits for its participants by addressing exercise barriers and facilitators using behaviour change techniques (BCTs), delivered in the Exercise and Educate model. This model is based on the COM-B behaviour change framework, a component of a larger behaviour change model referred to as the behaviour change wheel [43]. This framework identifies three necessary components for a behaviour to occur, capability (C), opportunity (O), and motivation (M), in which all components and the interactions between them contribute to a behaviour (B). Within ACE, the COM-B framework outlines the mechanisms through which exercise behaviour change can occur [43], including through exercise barriers and facilitators. The ACE programme has

pre-existing, built-in instructor fidelity checks conducted on an annual basis. Additionally, checklists are implemented to ensure that instructors are discussing scheduled BCTs on a weekly basis.

ACE has been offered in-person since early 2017, until the COVID-19 pandemic necessitated the rapid transition of the programme to an online platform in April 2020. This transition of the ACE programme to an online delivery format presented a unique opportunity to provide valuable, practice-based evidence that has the potential to direct future research and inform the safe and effective delivery of online exercise oncology programmes. The purpose of this project was to thus gather perspectives of people living with and beyond cancer who have experienced the transition from the in-person to online ACE programme. Specifically, understanding participants' preferences, facilitators, barriers, and experiences within the ACE programme will further our understanding of how to optimize online exercise oncology programme delivery. We hypothesised that (1) exercise barriers and facilitators for in-person and online exercise programmes will exist in both modes, but the type of barriers and facilitators will be different depending on the programme delivery mode, and that (2) less reported experience with or use of BCTs will be associated with more barriers, less facilitators, and lower exercise levels.

## 2. Materials and Methods

### 2.1. Participants

All participants were part of the larger ACE programme in Southern Alberta, who had transitioned from in-person to online (maintenance) programmes. New ACE participants who joined the programme during COVID-19 were not included in this study. Participants were invited via email to complete a single survey and an optional interview, conducted online via Survey Monkey and Zoom, respectively.

### 2.2. Study Design

An explanatory sequential mixed-methods design was used, following a pragmatist approach [44]. Pragmatism is a philosophical approach focused on finding solutions to practical problems using a variety of perspectives and methods and is commonly used in mixed methods research. This design was conducted in two phases. Quantitative data collection and analysis were conducted first, followed by qualitative data collection and analysis. This design allowed the exploration of qualitative results to expand upon and further understand the quantitative findings, as well as allowing the quantitative results to guide the purposive sampling for qualitative data collection [44].

Several strategies were employed in an attempt to enhance the validity and rigour of the qualitative data, including reflexivity and credibility [45]. To ensure reflexivity, multiple researchers took part in the interview data analysis, including the grouping and naming of themes. Generated themes were also compared back to the original data to confirm the thematic descriptions that we created were grounded in the data. Credibility was ensured via an effort to thoroughly describe each step of the research and data analysis processes to provide the reader with the ability to judge the generalizability and limitations of the outcomes.

### 2.3. Measures

#### 2.3.1. Demographics

Participants self-reported demographic information including name, date of birth, marital status, education level, annual family income, employment status, cancer diagnosis, start date of ACE baseline programme participation, self-identified gender, and self-identified race.

#### 2.3.2. Exercise Levels: Self-Report

The modified Godin Leisure-Time Exercise Questionnaire (GLTEQ) was used as a measure of subjective exercise levels [46]. The questionnaire asks participants to identify

their frequency and duration over the last week of four physical activity categories: light activity, moderate activity, strenuous activity, and resistance training. The GLTEQ allows for a total calculation of physical activity, MVPA, and levels of resistance activity on a weekly basis. The GLTEQ results were converted into scores for MVPA (to include moderate and strenuous exercise), resistance training (to include only resistance exercise), and MVPA plus resistance training, to determine the percentage of participants who were currently meeting cancer survivor guideline activity recommendations [47]. At the time of survey completion, each participant would have been in the maintenance phase of the ACE programme.

### 2.3.3. Exercise Levels: ACE Class Attendance

Attendance data for participants' most recent in-person (baseline or maintenance) exercise oncology programme (dates ranged from Winter 2017 to Winter 2020) and online (maintenance only) exercise programme (Summer 2020 to Winter 2021) were collected.

### 2.3.4. Exercise Setting Preferences

Participants were asked to indicate one of the following as their preferred exercise setting: online, in-person, or no preference. Participants were given the option to provide reasons for their exercise setting preference in a comment box.

### 2.3.5. Exercise Barriers and Facilitators

To measure participants' barriers and facilitators to exercise, the modified version of the Exercise Barriers/Benefits Scale (EBBS) was used [48]. EBBS scores were used to identify barriers to and benefits (facilitators) of exercise in online and in-person settings. EBBS is a 43-item, 4-point Likert scale ('strongly agree', 'agree', 'disagree', and 'strongly disagree') questionnaire that has demonstrated reliability and validity [49–51]. EBBS question prompts were adapted to cancer populations. The option 'not applicable' was added to the items 'I will prevent heart attacks by exercising', 'exercising will keep me from having high blood pressure', and 'my spouse (or significant other) does not encourage exercising'. Additionally, one question that could elicit negative feelings was removed: 'I will live longer if I exercise'. Therefore, the total EBBS score ranged from 39 to 168.

### 2.3.6. Behaviour Change Techniques

To measure participants' experiences with behaviour change techniques (BCTs), they were asked to report their BCT use and frequency during their in-person and online ACE programmes. In both settings, participants were specifically asked to identify if they used or engaged with a particular BCT (i.e., goal setting and social support). If a BCT was used, participants were further asked how often they used the BCT during the duration of either programme session. Eight BCTs in total were included in the survey, based upon BCTs incorporated into ACE programme delivery and instruction. Five were derived from the ACE 'Exercise and Educate' model: 1. principles of exercise and cancer, 2. goal setting, 3. behaviour change, relapse prevention, and motivation, 4. stress management and fatigue, and 5. social support and long-term maintenance. Additional BCTs that are commonly used within ACE were also evaluated, including the following: 1. verbal persuasion to boost self-efficacy, 2. providing feedback on performance, and 3. prompting review of behavioural and outcome goals.

### 2.3.7. Interviews

Individuals who confirmed willingness to participate in an interview and who were subsequently selected through purposive sampling took part. Purposive sampling for interviews considered age, gender, cancer diagnosis, and exercise setting preference. This purposive sampling was used to achieve variety across the sample in an attempt to hear varied perspectives [52]. Interview guides were created to foster open-ended responses about the survey topics to obtain a more detailed and contextualised understanding of participant experiences with barriers, facilitators, and BCTs in both delivery modes. Guides

were pilot tested with fellow research team members. Interviews were conducted via Zoom by author D.D., a female graduate student who received study-specific training, had recently completed a graduate-level qualitative research methodology course, and had semi-structured interview research training and experience. The interviewer had previous relationships of varying degrees with the participants via her role as an exercise instructor with the ACE programme, ensuring an initial level of rapport prior to interview conduction. All participants were aware of D.D.'s role as an ACE instructor prior to study commencement and that this research was conducted as part of her master's degree thesis project. All interviews were conducted in a one-on-one setting, with no other participants or researchers present. All interviews were audio-recorded and no repeat interviews were conducted. Some notes were hand-written by D.D. during the interviews and transcripts were not returned to participants for comment or correction.

### 2.4. Statistical Analysis

All quantitative analyses were conducted using SPSS 26 and Microsoft Excel v16.46. Results from the survey were summarised using descriptive statistics. To test our hypotheses, correlation models were used to evaluate the relationships between (1) Exercise Benefits and Barriers Scale (EBBS) scores and ACE class attendance, (2) EBBS scores and BCT use, and (3) BCT use and ACE class attendance in both delivery modes. Independent *t*-tests assessed differences in BCT use, EBBS scores, and ACE class attendance between delivery modes. Significance level was set at $p < 0.05$ [53].

### 2.5. Qualitative Analysis

Content analysis was used to analyse open-ended questions posed on the survey, using Hsieh and Shannon's conventional content analysis approach [54]. This type of content analysis uses an inductive approach to code ideas relevant to the research questions. Interviews were transcribed verbatim by the first author (DD) and subsequently analysed using guidelines for thematic analysis [55]. Consistent with thematic analysis guidelines and its underlying constructivist philosophy, the first author, who had previous experience with this methodological approach, conducted the analysis, with feedback and input from the rest of the research team [56]. Each transcript was read multiple times to familiarize the first author with the overall meaning of the data. Initial codes were generated using NVivo 12.0 software (QSR International) by inductively identifying and labelling ideas in the transcript related to the research questions. Codes with similar ideas were grouped and themes were identified and named. NCR, MM, and WB provided feedback on the grouping and naming of themes in an iterative process. Once final themes were identified, DD reread the transcripts and reviewed the themes to ensure that the data was coded. A description of each theme was then written, incorporating illustrative quotations. Participants did not provide feedback on the findings. The consolidated criteria for reporting qualitative research (COREQ) checklist was followed to help ensure clear and thorough reporting [57] (see Table S1).

## 3. Results

### 3.1. Survey Results

3.1.1. Participant Demographics

For the survey, 124 ACE participants who had taken part in both in-person and online ACE classes were identified and contacted via email. Of those contacted, 61 opened the survey and 57 provided complete data sets (46% response rate, 93% completion rate). No reasons were provided by participants for non-participation.

Most participants who completed the survey were female (84%) and had breast cancer (60%), reflective of the demographics of the larger ACE programme. The age of participants ranged from 44 to 84 years (mean ± SD; 62 ± 9 years) at the time of survey completion. The average time since date of diagnosis ranged from 1.3 to 27.4 years (mean ± SD; 5 ± 5 years and the average time since participants' baseline session start date ranged from

1.1 to 4.1 years (mean $\pm$ SD; 2.4 $\pm$ 1.1 years). A complete overview of survey participant demographics and clinical characteristics (i.e., cancer type) can be found in Table 1.

**Table 1.** Clinical characteristics and demographics of survey participants.

| Clinical Characteristic | No. of Participants |
|---|---|
| Gender [a] | |
| Male | 8 (14.0%) |
| Female | 48 (84.2%) |
| Another | 1 (1.8%) |
| Primary cancer type | |
| Breast cancer | 34 (59.6%) |
| Leukaemia or lymphoma | 7 (12.3%) |
| Multiple myeloma | 3 (5. 3%) |
| Prostate cancer | 2 (3.5%) |
| Lung cancer | 2 (3.5%) |
| Endometrial cancer | 2 (3.5%) |
| Multiple cancers | 2 (3.5%) |
| Colon cancer | 1 (1.8%) |
| Ovarian cancer | 1 (1.8%) |
| Thymus cancer | 1 (1.8%) |
| No cancer (support person) | 2 (3.5%) |
| Demographic variable | No. of participants |
| Race [a] | |
| Caucasian or white | 43 (75.4%) |
| White Anglo-Saxon Protestants | 3 (5.3%) |
| Chinese | 2 (3.5%) |
| Did not specify | 2 (3.5%) |
| Italian | 1 (1.8%) |
| German | 1 (1.8%) |
| Black | 1 (1.8%) |
| Eurasian | 1 (1.8%) |
| Mixed | 1 (1.8%) |
| Oriental | 1 (1.8%) |
| Arab | 1 (1.8%) |
| Employment Status | |
| Full-time | 10 (17.5%) |
| Retired | 29 (50.9%) |
| Homemaker | 3 (5.3%) |
| Part-time | 6 (10.5%) |
| Temporarily unemployed | 0 (0.0%) |
| Temporarily unemployed due to COVID-19 | 2 (3.5%) |
| Disability/sick leave | 7 (12.3%) |
| Student | 0 (0.0%) |
| Annual family income, CDN | |
| <CDN 20,000 | 0 (0.0%) |
| CDN 20,000–39,999 | 3 (5.3%) |
| CDN 40,000–59,999 | 3 (5.3%) |
| CDN 60,000–79,999 | 9 (15.8%) |
| >CDN 80,000 | 20 (35.1%) |
| Prefer not to specify | 22 (38.6%) |
| Education level | |
| Some high school | 0 (0.0%) |
| Completed high school | 2 (3.5%) |
| Some university/college | 4 (7.0%) |
| Completed university/college | 34 (59.6%) |
| Some graduate school | 3 (5.3%) |
| Completed graduate school | 14 (24.6%) |
| Marital Status | |
| Never married | 1 (1.8%) |
| Married | 41 (71.9%) |
| Common law | 4 (7.0%) |
| Separated | 1 (1.8%) |
| Widowed | 7 (12.3%) |
| Divorced | 3 (5.3%) |

SD, standard deviation; ACE, Alberta Cancer Exercise. [a] Race and gender demographic variables were self-identified through an open-ended question.

### 3.1.2. Exercise Levels: Self-Report

The average MVPA was $186 \pm 169$ min/week. A total of 34 participants (60%) met both the MVPA and resistance training recommendations.

### 3.1.3. Exercise Levels: ACE Class Attendance

Attendance data were pulled from survey respondents' most recent in-person and online ACE classes. In-person class attendance was either from participants' baseline class ($n = 27$, 52%) or maintenance class ($n = 25$, 48%). In-person class attendance data could not be located for some participants due to limited access to some fitness facilities because of COVID-19 ($n = 5$). All online class attendance was taken from maintenance classes. The average for online attendance, taken from the most recent programme offering, was 79.8% of total classes attended. For participants' most recent in-person classes, average attendance was 76.8%. No statistically significant differences were identified between in-person and online attendance data.

### 3.1.4. Exercise Delivery Mode Preferences

Survey data indicated a preference for the in-person ACE maintenance classes, with 33 participants (57.9%), followed by 18 (31.6%) preferring online, while 6 (10.5%) indicated no preference, or ambivalence. The most cited reasons for preferring the in-person classes were social interaction, equipment, safety, and the ability to receive more tailored, one-on-one feedback from instructors. For the online classes, the most common reasons for preference included diminished commuting-related factors, convenience, improved confidence when exercising, and decreased fatigue. For those who were ambivalent, reasons for enjoying the in-person programme included social interaction and safety, whereas reasons for appreciating the online classes included convenience. However, online classes were noted to not provide an equivalent level of social support that was available in-person, including a diminished ability to get to know other participants on a personal level.

### 3.1.5. Exercise Barriers and Facilitators

EBBS scores between in-person and online exercise classes indicated significantly fewer barriers ($p < 0.01$) but also fewer benefits (facilitators) ($p < 0.01$) within the online delivery mode (see Table 2). The most valued facilitators (i.e., response of 'strongly agree') for the in-person exercise delivery mode were 'exercising improves my mental health' ($n = 44/57$, 77%), 'exercising lets me have contact with friends and people I enjoy' ($n = 38/57$, 67%), and 'exercising is a good way for me to meet new people' ($n = 27/57$, 47%). Barriers that were less prevalent (i.e., response of 'strongly disagree') in the online exercise delivery mode were 'exercising takes too much of my time' (58% online vs. 40% in-person), 'places for me to exercise are too far away' (79% online vs. 26% in-person), 'there are few too places for me to exercise' (47% online vs. 35% in-person), and 'I am too embarrassed to exercise' (65% online vs. 53% in-person).

**Table 2.** Average Exercise Benefits and Barrier Scale (EBBS) scores: total, benefit, and barrier scores and differences between in-person to online.

|  | **In-Person Score: Mean $\pm$ SD** | **Online Score: Mean $\pm$ SD** | ***p* Value** |
|---|---|---|---|
| EBBS total | $138.02 \pm 14.29$ | $137.05 \pm 13.99$ | $p = 0.35$ |
| Benefits | $94.23 \pm 10.69$ | $91.16 \pm 10.96$ | $p = 0.00$ |
| Barriers | $25.16 \pm 5.13$ | $23.05 \pm 4.31$ | $p = 0.00$ |

Note. Two-tailed, paired *t*-tests were conducted. SD, standard deviation. The term 'benefits' in this scale is used synonymously with facilitators.

### 3.1.6. Behaviour Change Technique Support

The average number of BCTs each participant indicated using or receiving from their instructors was significantly lower ($p < 0.01$) in the online than in the in-person environment, with an average of 5.5 (out of 8) BCTs reported in-person and 4.6 reported online. There

was a trend toward greater use of BCTs in in-person classes as compared to online classes ($p = 0.065$). The BCTs with the largest differences between delivery modes included social support, education on exercising with a cancer diagnosis, creating consistent exercise habits, and stress management and fatigue. In all cases, the in-person setting showed to be more conducive to BCT use based on participants' reporting of the peer social support experience and more direct instructor feedback that they received. They expressed that this enabled them to exert more effort during class time as well as boosted their motivation to attend. BCTs, based on the "Exercise and Educate" model within ACE, including providing feedback on performance, verbal persuasion to boost self-efficacy, and goal setting were similar across settings.

Correlation analyses were conducted evaluating potential relationships between EEBS scores, BCTs, and attendance data in either in-person or online classes. No significant relationships between these three variables were found in either exercise delivery mode (see Table 3).

**Table 3.** Correlation analyses for EBBS Scores, BCTs, and attendance data.

| | Online Delivery Mode | | |
| --- | --- | --- | --- |
| | EBBS score | Number of BCTs used | % of classes attended |
| EBBS score | - | | |
| Number of BCTs used | 0.12 | - | |
| % of classes attended | 0.02 | 0.15 | - |
| | In-Person Delivery Mode | | |
| | EBBS score | Number of BCTs used | % of classes attended |
| EBBS Score | - | | |
| Number of BCTs used | 0.09 | - | |
| % of classes attended | -0.20 | 0.23 | - |

Note. Values represent correlation coefficients representing relationships between identified variables.

*3.2. Interviews*

Nearly all survey participants (53/57, 93%) agreed to be contacted for an interview. A total of 21 participants were purposively sampled and contacted via email to participate in an interview, with 19 responding and completing interviews (91% RR; 100% CR). The average interview length was 45 min and 43 s (a range of 31:49 to 68:03 min). The majority of participants who completed an interview were female (68%), had breast cancer (37%), identified as white (84%), were retired (68%), married (74%), and were older adults (with an average age of 63 years). While acknowledging the importance of all the themes generated from the participants' knowledge, given the focus of this research, the results and discussion centre on participant experiences with the transition from in-person to online delivery, and similar or differing experiences in both settings. Additional themes that were more relevant to the overall ACE programme, satisfaction with the programme, and future programme offerings or improvements are not included. These themes may be explored in future research and incorporated into quality improvement cycles for ACE and other exercise oncology programmes.

Thematic Analysis

The four themes generated through the thematic analysis are described below.

- Theme 1: It's been the best route that we could take, given the circumstances.

Participants described feeling isolated during COVID-19 and how ACE transitioning to an online format allowed them to continue to exercise and reap the physical and social benefits of ACE that they had experienced in in-person classes. These physical and social benefits were a key component to participants' mental and physical health throughout the pandemic.

"Initially, when the lockdown came along, I thought [I was] going to lose all these things that are keeping me from losing my mind. So, when the ACE program [went] online, [. . .] I was just so happy and relieved [. . .]. It's been such an important way for me to feel like I'm connecting with other human beings during the day [. . .]. That's been kind of the guiding light for my mental [. . .]." (P42)

Despite participants expressing gratitude for the opportunity to continue with ACE during the pandemic, individual differences were still reflected in exercise barriers and/or facilitators. One key factor to a successful transition frequently noted by participants was having prior experience with the in-person classes.

"Had it been proposed me as online straight off, I probably would have passed [. . .]. I tend to think if it's physical, then I need someone else there with me [. . .]. But I think the fact that I was already in the [ACE] system meant that [. . .] I'll try it. If it doesn't work, I'll just move on." (P77)

Despite the online classes not being seen as equivalent to the in-person classes by some participants in terms of the social support and personalised feedback provided by instructors, the ability to continue to see others and to reap the benefits of continued instruction from exercise experts was described as important.

"It's not quite the same interaction, because Zoom's one person at a time [. . .], you don't get the same type of conversation. But there's that opportunity to ask questions and have discussion [. . .]. So, for a lot of classes, people are logging in early and then there's some socialization [. . .]. [. . .] [Having] live instructors, real time, and adapting as you go, that's probably been the best route that we could take, given the circumstances." (P56)

- Theme 2: The online environment improved my experience with exercise in some ways, but made it more challenging in others.

This theme captures the varied experiences of participants while exercising with ACE in the online environment. Despite the attempts made by the ACE team to create a beneficial environment online, participants noted that generally they received fewer physical and social benefits in the online environment. The decrease in benefits was often attributed to inherent limitations associated with exercising using the online programme.

"What I miss about the in-person is [the instructors] don't really have the ability online to walk around and check on us [. . .]. [. . .] it was easier to get that kind of that kind of help one-on-one. When [. . .] I'm in a square [on] Zoom, it's difficult to give that kind of help [. . .]. And it's a bit more difficult to get a really good handle on what people's limitations are when it's online." (P77)

However, some participants noted an increase in encouragement from instructors and physical benefits online.

"The positive reinforcement that's given by the moderators and the instructors has taken on a whole new dimension [. . .]. Because the instructors are more focused on your form and structure [. . .]. I would say the encouragement [online] is more affordable now than it was before [in-person]." (P18)

Despite instructors' best efforts to encourage social connections between participants, limitations still existed for fostering personal relationships between participants.

"You get a chance to know the other people a little bit more [in-person] [. . .]. As opposed to online, other than the people I was in the classroom [in-person] with, I don't know anything about these other 10 people." (P6)

Despite potential limitations to social benefits, participants noted other benefits of exercising using the online programme. New benefits or facilitators included more time throughout the day, less exacerbation of fatigue symptoms, and an increased level of confidence while exercising due to the comfort afforded by attending in the home environment.

"One benefit of [online] is that it's way less easy for me to talk myself out of a class [...]. When you have to physically leave your house and drive somewhere, on the days when I'm feeling a little bit low, it's much easier for me to [attend online] [...]. (P42)

"[...] what I'm learning now is [exercising using the online program is] giving me the confidence [...] to listen to my body to do what I need to do [...]. [Because] You're watching me, but I'm more alone. I do sometimes try new things. I think I'm less intimidated." (P75)

In addition, participants described less barriers to attending the online classes, including no commute time and no need to walk or drive in poor weather. Motivation to attend classes was potentially both increased and decreased across participants by these factors.

"The accessibility, especially when it's 20, 30 below, so much easier to be motivated to go online and do a program than it is to get bundled up and walk [...]." (P2)

"And [...] you just didn't feel like you had the same incentive to attend when it just meant going upstairs as opposed to preparing to go somewhere." (P6)

Lastly, this theme describes participant experiences with BCTs in the online classes, which some participants described as being similar or slightly less prevalent online.

"The education piece [helps my exercise habits]. And having [the instructors] individually educate me on proper technique to get the benefit. So even though it's difficult via Zoom, it still happens [...]. It still modifies the behaviour; it still creates that desire [...]. I'm still learning new exercises." (P56)

- Theme 3: My in-person ACE experience afforded many benefits, but I still faced barriers to attending.

This theme captures the varied experiences of participants while exercising with ACE in the in-person environment. One of the most important aspects described of exercising in-person were the social benefits received while exercising. Some participants described the social benefits derived from the class as a 'bonus' as opposed to an essential component of ACE, while others felt that the social interaction in-person was the best part of ACE and struggled to attend online.

"[...] by quite a wide margin, my preference would be in-person. Because of the value to me of some sort of social contact. And as a result, the sense of community connection, [...] the sense of safety that comes from the instructor telling you how to do it right, [and] the presence of a large number of people in the room." (P18)

"I think the social support thing is more important to some people than others. I'm lucky, I've got a really strong support system. And if we weren't able to do any more in-person classes forever, I'd still be okay [...]. That one-to-one and the help when I needed it. And a couple of good friends that I've made. Those were all bonuses." (P16)

Despite the beneficial social support that occurred in-person, attending classes in this delivery mode regularly was still difficult for some participants. Barriers to in-person classes included poor weather, commute time (walking or driving), exacerbated fatigue, and parking costs in-person.

"When it was a cold and wintery and slippery day [...] by the time I got ready, drove through the weather, and parked and walked to the university. I'm like, whew, okay, I think I'll just go back. That part of it is easier being at home [...]. It was a bit challenging to do that walk." (P16)

Some people felt as if the social support aspect and the benefit of interacting with the instructors in-person was worth combatting these barriers to come in-person, whereas

others felt that the convenience of the online programme was superior to the social support received in-person.

> "But [...] it didn't matter what the weather was like, [I] still showed up [to see others in the class]." (P6)

> "While I enjoy the social support and the interaction, [...] the social aspect for me isn't a massive thing [...]. But in terms of reduction of barriers [online], that I did find was really high, because we don't have the commute time, it was much easier to interweave it and fit it into the day [...]. I found a lot of the barriers to regular exercise actually did drop for me." (P109)

This theme also describes participant experiences with BCTs in the in-person classes, which largely surrounded the social support benefits derived from in-person, the education received on exercising, and the feedback and encouragement from instructors, which were generally described as being more prevalent in the in-person environment.

> "Obviously, [behaviour change techniques are] better in-person than they are online. Especially when [the instructors] have 20 people [in class on Zoom], [...] that's a lot. I've actually noticed the difference. Because even with the 12 people, [...] somebody would be saying oh, that's good, [NAME], keep that up. But now, [online], it's not very often that you hear that." (P25)

- Theme 4: My goal is to have a good quality of life and maintain my level of functionality through moving more.

This theme captured when participants spoke about their overarching goals of maintaining a good quality of life and how they did not feel they needed 'other skills' to engage in exercise. For example, participants described that they did not focus on setting specific physical goals that they need to achieve. Instead, for them, being generally active was their goal in order to maintain healthy physical functioning.

> "My goal, if you want to call it that, is to do the exercises, as best as I can, and hopefully better than I did them the last time. [...] It's just keeping my body moving and functioning properly that's important to me [...]. I don't set an exercise goal, per se, [...] some days even showing up is a challenge in itself." (P39)

Ultimately, participants described a wide variety of experiences with the in-person and online classes. These delivery modes had a variable impact on participant barriers and facilitators and experience with BCTs, ultimately leading to variable exercise delivery mode preferences across participants. The impact of their cancer diagnoses, other factors in their lives (including the impact of COVID-19), and where they were along the treatment trajectory—these all influenced the perspectives, participation, and experiences of ACE participants.

## 4. Discussion

The current work examined the differences between in-person and online exercise oncology maintenance programme delivery on participant perceived barriers, facilitators, transitions, and experiences with BCTs. Both barriers and facilitators were perceived as being higher in the in-person class environment. Additionally, BCTs were viewed as being used more frequently and as more impactful within the in-person environment. Given the potential role of the online delivery of exercise oncology as a supportive cancer care resource, this work is critical for understanding the differences in these two delivery modes, so that intervention effectiveness can ultimately be optimised. The lessons learned from this research can be applied to future online exercise oncology offerings, which have the potential to reach more individuals living with and beyond cancer, particularly those who cannot typically access in-person classes, such as those living in rural and remote locations.

In our study, similar attendance results were found between in-person and online delivery modes. This relatively high engagement in the online delivery mode may have been due

to the elements of ACE maintenance delivery, including the synchronous, supervised, and group-based nature of the online programme, that facilitated greater engagement and supported building social connections. This is consistent with past interventions that demonstrate higher engagement and attendance through videoconferencing [26,28,30,31,58,59]. It is, however, important to consider the order in which participants attended the different ACE exercise environments. All participants in this research first took part in in-person classes, followed by online classes. Our results may have varied if participants experienced online ACE programming prior to in-person participation. Calls for telehealth interventions that deliver synchronous, supervised, and group-based exercise sessions, similar to what has been carried out for in-person exercise programmes for individuals living with cancer, have been made and will be important to further examine [19,39].

Findings from this research can be applied to optimize online delivery for exercise oncology programmes. First, the current results reinforce that social support and a sense of community need to be continually fostered in the online exercise oncology environment. Ultimately, the group-based aspect of the ACE programme was seen as a facilitator for attending it on a regular basis in both delivery modes. However, participants also described that receiving the 'usual' level of support (social, BCTs such as goal setting and instructor feedback) as in-person was difficult in the online environment. This was largely attributed to an inability to have informal, one-on-one conversations via an online platform, limiting opportunities for social interactions, similar to other studies [37]. This potential lack of support from simply chatting with other participants was an important facilitator of exercise attendance for in-person classes that was not as available in the online ACE setting. In our exercise oncology programmes, we provide social support in the online environment by facilitating pre- and post-exercise Zoom chats (15 min each) on topics brought up by participants. Further understanding how these additional times to connect may bolster positive outcomes and support exercise adherence will be essential for optimising the design, delivery, and impact of future exercise oncology programmes. Looking towards the future, the development of videoconferencing technology that allows one-on-one conversations or more flexible breakout rooms (i.e., one-on-one or small group conversations permitted without having to leave the main Zoom room) may bolster the success of utilising videoconferencing as a tool for exercise, and exercise oncology specifically. Exploring other methods of bolstering social support in the online environment will be a key component to consider when attempting to deliver, design, or improve future exercise oncology programmes.

Second, more steps need to be taken to provide participants with feedback on exercise techniques to ensure safety and optimize potential physical programme benefits. One advantage to synchronous delivery is increasing the potential to deliver interventions via telecommunication technologies with higher levels of participant engagement and safety, compared to those with asynchronous delivery [28,39,60]. This is consistent with our results describing the benefits of synchronous supervision and the ability to receive immediate exercise modifications online. The safety benefits derived from constant supervision from exercise oncology experts can be applied to other online exercise programmes. In addition, ACE facilitated safety and effectiveness via weekly pre-class communication emails containing the class exercise plan, the Zoom link and password, and a list of necessary equipment needed for each class. Participants could then ask questions prior to or at the start of a class on specific exercise modifications or ways to challenge themselves during class time. Participants described these emails as fostering a more open line of communication between participants and instructors and as allowing them to feel more prepared to exercise each class. This was noted as a key benefit enhancing feelings of readiness for exercising online.

Third, evaluating the cost of designing and offering online exercise oncology programmes is an integral part of building a sustainable programme that can continue to help cancer survivors adhere to consistent exercise, and built a habit of moving more. For participants, offering online programming limits the cost of attendance, negating the need for travel or parking costs [1]. For providers, examinations of cost analyses in future

research will be essential to enhance sustainability [1,40]. mHealth and other telehealth interventions have already demonstrated the potential to decrease the cost of providing complementary healthcare services such as exercise [26–28,31]. A call for the future examination of cost-effective interventions demonstrating real-world feasibility and applicability has been made and will be crucial moving forward [32].

Recent work and reviews evaluating the challenges facing implementing telehealth technology have highlighted recommendations for future researchers to enhance safety and effectiveness through various strategies. Recommendations have included using pre-class questionnaires to identify daily symptoms and challenges for each participant, providing participants with adequate instructions on how to use the web-based platform of choice, creating group-based environments to foster social support, considering privacy and protection considerations while conducting classes using an online platform, and fostering a rapport with participants via getting to know participants individually early on in the programme and tailoring feedback on an individual level, among others [39,61,62]. In our current exercise online delivery study for rural and remote individuals living with and beyond cancer [63,64], these factors, as well as ensuring participants exercise with their cameras on, documenting each participant's address where they are exercising in case of a need to direct emergency services to them, and adding a moderator for each class, who acts as a support person for the instructor to further ensure the safety of all participants, are being implemented to enhance both safety and potential effectiveness of online exercise delivery.

Last, the wide variety of findings highlights the uniqueness of every individual living with cancer. In the field of exercise oncology, a variety of factors need to be considered to deliver the most effective exercise experience possible to everyone. This was also seen in the individuality in experiences with BCTs, exercise barriers and facilitators, and exercise delivery mode preferences. This variety in the multiple aspects that contribute towards the success and uptake of an exercise programme highlight the need for tailoring and individualisation within an exercise programme. This tailored approach is essential for successful implementation and requires moving beyond a generic 'one-size-fits-all' exercise prescription.

## 5. Strengths and Limitations

This study had some notable strengths, including rich data derived from both quantitative and qualitative data collection methods and a strong response rate for surveys (46%) and interviews (91%). Additionally, the interviewer (DD) had prior personal relationships with a majority of the interview participants (16/19, 84%), leveraging an already established rapport to generate meaningful and candid qualitative data.

Limitations to this work included only collecting data at one point in time. This may have introduced a recall bias when prompting participants to recall their barriers, facilitators, and BCT use in the-person exercise delivery mode. Second, the sample of ACE maintenance participants were more active than is the general cancer population. Therefore, the insights and experiences of this population may not be reflective of a less active population of people living with cancer. Third, the sample for this study and the ACE population tend to be middle- to upper-class, white, highly educated, and retired. This population may have potentially been subject to less negative effects from the pandemic on their overall well-being and had greater access to the technology necessary to join online programming. Therefore, the results from this study may not be generalizable to all individuals living with cancer who were engaging in exercise during or beyond the pandemic. Another limitation of note that has been identified in other exercise oncology research [65,66] is that the majority of our study population were females with breast cancer, which is representative of the larger ACE participant data to date. Future studies should attempt to recruit participants representing a more diverse sample of cancer diagnoses. For instance, the EXercise for Cancer to Enhance Living Well (EXCEL) project reaches rural and remote people living with and beyond cancer, and has created tumour-specific (i.e., lung,

head-and-neck, and neurology specific cancer) resources and programmes to encourage participation and safety for various tumour groups [63,64,67]. Programmes such as these may be an impactful way to reach a more diverse population of people living with and beyond cancer.

## 6. Conclusions and Directions for Future Research

The present study indicates that ACE participants experience a range of barriers and facilitators to both in-person and online exercise oncology delivery modes. Despite a decrease in both barriers and facilitators in the online class environment, attendance of ACE maintenance classes online remained the same as that for the in-person delivery mode. Ultimately, participants felt fortunate to have continued access to ACE during the pandemic to keep them active and connected to others. Future directions for this research are necessary to ensure individuals living with cancer remain supported as complementary therapy services, such as exercise, remain out of reach for many due to the COVID-19 pandemic, and potentially beyond. The results of this research will remain relevant as we aim to increase the reach of exercise oncology programming to underserved populations of individuals living with cancer (i.e., rural/remote, immunocompromised, and young adult populations) by utilising synchronous, supervised, and group-based telehealth exercise oncology programmes.

**Supplementary Materials:** The following supporting information can be downloaded at: https://www.mdpi.com/article/10.3390/curroncol30080534/s1, Table S1: COREQ (COnsolidated criteria for REporting Qualitative research) Checklist.

**Author Contributions:** Conceptualisation, D.D. and S.N.C.-R.; methodology, D.D., S.N.C.-R., M.H.M. and W.B.; software, D.D.; validation, S.N.C.-R., M.H.M. and W.B.; formal analysis, D.D.; investigation, D.D.; resources, D.D. and S.N.C.-R.; data curation, D.D.; writing—original draft preparation, D.D.; writing—review and editing, S.N.C.-R., M.H.M., W.B. and M.L.M.; visualisation, D.D. and S.N.C.-R.; supervision, S.N.C.-R.; project administration, S.N.C.-R. and M.L.M. All authors have read and agreed to the published version of the manuscript.

**Funding:** This research received no external funding.

**Institutional Review Board Statement:** All procedures performed in studies involving human participants were in accordance with the ethical standards of the University of Calgary Health Research Ethics Board of Alberta—Cancer Committee (HREBA.CC-20-0379) and with the 1964 Helsinki declaration and its later amendments or comparable ethical standards.

**Informed Consent Statement:** Informed consent was obtained from all subjects involved in the study.

**Data Availability Statement:** Restrictions apply to the availability of these data. Data were obtained from the Alberta Cancer Exercise project and are available from the authors with the permission of the Alberta Cancer Exercise project.

**Acknowledgments:** The authors thank the participants who contributed to this study and the Alberta Cancer Exercise team who contributed to this research project. The authors would also like to thank the Alberta Cancer Foundation for financially supporting the Alberta Cancer Exercise study.

**Conflicts of Interest:** The authors declare no conflict of interest.

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
