# Peer review of "Understanding In-Person and Online Exercise Oncology Programme Delivery: A Mixed-Methods Approach to Participant Perspectives"

_curroncol, doi:10.3390/curroncol30080534_

Round 1

Reviewer 1 Report

The manuscript by Duchek et al. renders strength in that the authors evaluate a research question that has not previous been evaluated. I greatly appreciate the authors taking this journey so that programs like these can continue to reach more individuals in need. Please consider my comments below as feedback for making the manuscript more easily interpreted by the reader:

Specific Comments:

o   Line 21: “exercising online” may be confusing to the reader, perhaps “to exercise using the online program” or “digitally-delivered exercise program” or something similar.

o   Table 1: Is your male/female data possibly flipped give your n for breast cancer as the primary cancer type and the text above or it may all need to be shifted down on row to have a value for “another” rather than “Gender”? It would also be useful to include the total N in the table.

o   Line 189: A brief overview of the recommendations provided may help the provide some additional context to the reader.

o   Section 3.1.5: It might be more clear to the reader to include the most prevalent and least prevalent facilitators and barriers to each modality including  n, %?

o   Table 2: You may want to note somewhere that benefits = facilitators.

o   Line 226-229: It is unclear what the difference is between the two statements but two different p-values are noted?

o   Table 3: It is unclear what is presented in Table 3, please clarify further. Are these coefficients or p-values?

o   Discussion: Assuming I am understanding the program correctly, can the authors speculate on the carryover effects of starting in person and finishing online? How can/should this be considered for future design of programs?

Author Response

Please see the attachment. Note that all comments are included in one document as some comments are addressed in different responses to a separate reviewer.

Reviewer 2 Report

This is an interesting study that could have important implications for design and implementation of future telehealth and oncology exercise studies. However, as the results are currently presented, the study does not offer any new information about telehealth in oncology. A more indepth analysis of the qualitative data and how it triangulates with the quantitative data will help achieve this.

Specific comments are as follows

Title: please include study design in the title

Introduction.

Given the overall intent of using telehealth (I think!) is to broaden the reach of exercise via telehealth for cancer survivors broadly, less emphasis should be placed on COVID 19 in the introductory paragraph and more so on why exercise is important.

You have conducted a comprehensive literature review however, given the volume of telehealth research conducted since COVID-19, you need to further articulate the need for this particular study. That is, what is known and not known from a qualitative perspective and why this is important. For example, Dennett et al 2021 (JMIR Cancer) conducted a similar mixed methods study from an Australian perspective.

Given the aims specifically state the need to identify barriers and facilitators to exercise for both in person and telehealth, you need to briefly introduce known barriers and facilitators to exercise in oncology in general.

Methods

Participants – please briefly outline the inclusion criteria for participants of the ACE program

Even though you have provided a description of ACE in the introduction, it is unclear exactly what the model looked like with COVID. Given the process required in setting up telehealth, the transition to telehealth is usually more complicated than just changing the mode as there are other staffing and administrative things to consider. Please add an intervention section with the telehealth intervention detailed according to TiDier guidelines. If possible, it would be worth having this in a table to directly compare with the in person intervention

Please highlight if any reporting checklists were followed e.g. COREQ , CROSS

Given behavior change techniques were used, was anything else done to ensure the fidelity of sessions (e.g. a documentation audit)

Please describe the mode, duration and by whom the interviews were conducted

Briefly include whether any measures were taken to ensure the trustworthiness and reliability of qualitative data

Results

It is unclear whether exercise levels reported are before/after or during completion of the ACE intervention. Please specify. I wonder if part of this being missing is because patients may be longterm attenders whereas others may have just started the program. Therefore, if this is the case it may also be worth documenting how many new participants entered the program during CoVID

It is unclear to me how you know BCTs were more conducive to inperson? What do you exactly mean by this? Broader use of techniques? More frequency of one over the other?

Interview quotes are long and could be more concise with paraphrasing

A more indepth analysis of interview data is required. Theme 2 and 3 are basically the same, (about still facing barriers) so maybe there is an overall theme that barriers will exist regardless of the modality (note your first theme doesn’t really relate to the research question at a glance). A figure +/- table of themes would help articulate this better

Given this is a mixed methods study, a section relating to triangulation of the quantitative and qualitative data is required.

Discussion

The key findings could be made clearer and have more emphasis in the first paragraph to signpost the rest of the discussion.

Given the emphasis that this study could be useful in the future, there could be more emphasis on published implementation efforts in this field (and perhaps other chronic disease literature) along with some suggestions on how we can facilitate implementation of telehealth in the future.

Include a line on the study’s strengths – large pragmatic sample, mixed methods etc

The conclusion could be made more concise. The second paragraph could almost be removed.

Author Response

(The authors gave the same response as above.)

Reviewer 3 Report

The manuscript by Duchek et al. is a useful contribution to the literature on exercise in oncology. The Alberta Cancer Exercise program was forced to transition from an in person exercise program to an online program due to the barriers crated by Covid-19. It would be helpful to have a more fulsome description of the ACE including how the program is promoted, how potential participants are evaluated before entering the program and location of the fitness facilities. (I.e. are there fitness facilities within the cancer centres in Alberta or do patients have to travel to community centres or other health facilities?) With the transitioning to online, what would participants need to have in their home in order to participate in the exercise program and was that a barrier to lower SES individuals.  Also please describe briefly what instruction is provided to participants and whether it is individualized based on performance level?

The Exercise Barriers/Benefits Scale is said to have been modified for the cancer population. A number of items were deleted as "not applicable".  Were any cancer specific items  added?

Table 1 needs to be reformatted. The right-hand column is headed "No. of Participants " but the first three items relate to age, time since date of diagnosis and time since ACE baseline sessions started. It might be appropriate to either have a separate small table for these items or more simply exclude these details from the table and described them in the text. There also appears to be an error in the table as it indicates that 48 (84.2%) of participants were male. The text indicates that the majority of the participants were females with breast cancer. This appears to be a formatting issue.  

The interviews were undertaken in a rather small number of respondents (n =19) and although the demographics are provided,  when comments are included in the text, there is no indication as to whether it is from a male or female voice. The attitudes towards in person versus online may well differ by sex. The fact that there are relatively small numbers of patients in the study and they are predominantly female patients with breast cancer is one of the main limitations of the study as it limits the ability to generalize the findings, The authors might discuss why this seems to be a common problem in exercise studies in oncology and what they think could be done about it in their section on directions for future research.

Author Response

(The authors gave the same response as above.)
